# HIGHWAY AND RESIDUAL NETWORKS LEARN UNROLLED ITERATIVE ESTIMATION

**Klaus Greff**
The Swiss AI Lab IDSIA (USI-SUPSI)

**Rupesh K. Srivastava & Jürgen Schmidhuber**
The Swiss AI Lab IDSIA (USI-SUPSI) & NNAISENSE, Lugano, Switzerland
{klaus,rupesh,juergen}@idsia.ch

## ABSTRACT

The past year saw the introduction of new architectures such as Highway networks (Srivastava et al., 2015a) and Residual networks (He et al., 2015) which, for the first time, enabled the training of feedforward networks with dozens to hundreds of layers using simple gradient descent. While depth of representation has been posited as a primary reason for their success, there are indications that these architectures defy a popular view of deep learning as a hierarchical computation of increasingly abstract features at each layer.

In this report, we argue that this view is incomplete and does not adequately explain several recent findings. We propose an alternative viewpoint based on *unrolled iterative estimation*—a group of successive layers iteratively refine their estimates of the same features instead of computing an entirely new representation. We demonstrate that this viewpoint directly leads to the construction of Highway and Residual networks. Finally we provide preliminary experiments to discuss the similarities and differences between the two architectures.

## 1 INTRODUCTION

Deep learning can be thought of as learning many levels of representation of the input which form a hierarchy of concepts (Deng & Yu, 2014; Goodfellow et al., 2016; LeCun et al., 2015) (but note that this is not the only view: cf. Schmidhuber (2015)). With fixed computational budget, deeper architectures are believed to possess greater representational power and, consequently, higher performance than shallower models. Intuitively, each layer of a deep neural network computes a new level of representation. For convolutional networks, Zeiler & Fergus (2014) visualized the features computed by each layer, and demonstrated that they in fact become increasingly abstract with depth. We refer to this way of thinking about neural networks as the *representation view*, which probably dates back to Hubel & Wiesel (1962). The representation view links the layers in a network to the abstraction levels of their representations, and as such represents a pervasive assumption in many recent publications including He et al. (2015) who describe the success of their Residual networks

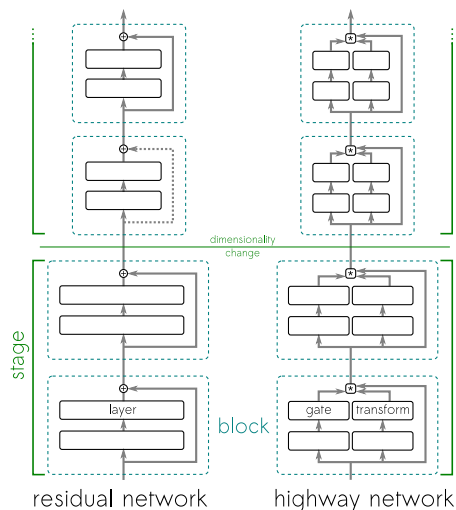

Figure 1: Illustrating our usage of *blocks* and *stages* in Highway and Residual networks.

like this: "Solely due to our extremely deep representations, we obtain a 28% relative improvement on the COCO object detection dataset."

Surprisingly, increasing the depth of a network beyond a certain point often leads to a decline in performance *even on the training set* (Srivastava et al., 2015a). Since adding more layers cannot decrease representational power, this phenomenon is usually attributed to the vanishing gradient problem (Hochreiter, 1991). Therefore, even though deeper models are more powerful in principle, they often fall short in practice.

Recently, training feedforward networks with hundreds of layers has become feasible through the invention of Highway networks Srivastava et al. (2015a) and Residual networks (ResNets; He et al. 2015). The latter have been widely successful in computer vision, advancing the state of the art on many benchmarks and winning several pattern recognition competitions (He et al., 2015), while Highway networks have been used to improve language modeling (Kim et al., 2015; Jozefowicz et al., 2016; Zilly et al., 2016) and translation (Lee et al., 2016). Both architectures have been introduced with the explicit goal of training deeper models.

There are, however, some surprising findings that seem to contradict the applicability of the representation view to these very deep networks. For example, it has been reported that removing almost any layer from a trained Highway or Residual network has only minimal effect on its overall performance (Srivastava et al., 2015b; Veit et al., 2016). This idea has been extended to a layerwise dropout as a regularizer for ResNets (Huang et al., 2016b). But if each layer supposedly builds a new level of representation from the previous one, then removing any layer should critically disrupt the input for the following layer. So how is it possible that doing so seems to have only a negligible effect on the network output? Veit et al. (2016) even demonstrated that shuffling some of the layers in a trained ResNet barely affects performance.

It has been argued that ResNets are better understood as ensembles of shallow networks (Huang et al., 2016b; Veit et al., 2016; Abdi & Nahavandi, 2016). According to this interpretation, ResNets implicitly average exponentially many subnetworks, each of which only use a subset of the layers. But the question remains open as to how a layer in such a subnetwork can successfully operate with changing input representations. This, along with other findings, begs the question as to whether the representation view is appropriate for understanding these new architectures.

In this paper, we propose a new interpretation that reconciles the representation view with the operation of Highway and Residual networks: functional *blocks*[1] in these networks **do not** compute entirely new representations; instead, they engage in an *unrolled iterative estimation* of representations that refine/improve upon their input representation, thus *preserving feature identity*. The transition to a new level of representation occurs when a dimensionality change—through projection—separates two groups of blocks which we refer to as a *stage* (Figure 1). Taking this perspective, we are able to explain previously elusive findings such as the effects of lesioning and shuffling. Furthermore, we formalize this notion and use it to directly derive Residual and Highway networks. Finally, we present some preliminary experiments to compare these two architectures and investigate some of their relative advantages and disadvantages.

## 2 CHALLENGING THE REPRESENTATION VIEW

This section provides a brief survey of some the findings and points of contention that seem to contradict a representation view of Highway and Residual networks.

**Staying Close to the Inputs.** The success of ResNets has been partly attributed to the fact that they obviate the need to learn the identity mapping, which is difficult. However, learning the negative identity (so that a feature can replaced by a higher level one) should be at least as difficult. The fact that the residual form is useful indicates that Residual blocks typically stay close to the input representation, rather than replacing it.

The analysis by Srivastava et al. (2015a) shows that in trained Highway networks, the activity of the transform gates is often sparse for each individual sample, while their average activity over all training samples is non-sparse. Most units learn to copy their inputs and only replace features selectively. Again, this means that most of the features are propagated unchanged rather than being combined

---

[1]We refer to the building blocks of a ResNet—a few layers with an identity skip connection—as a Residual block (He et al., 2015). Analogously, in a Highway network, we refer to a collection of layers with a gated skip connection as a Highway block. See Figure 1 for an illustration.

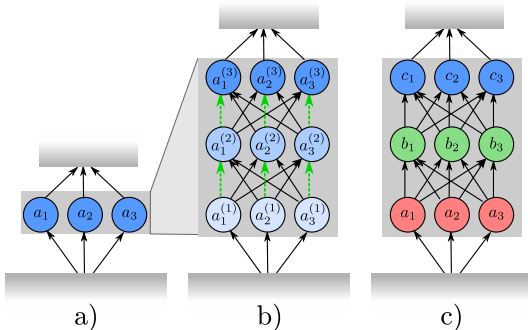

Figure 2: (a) A single neural network layer that directly computes the desired representation. (b) The unrolled iterative estimation stage (e.g. from a Residual network) stretches the computation over three layers by first providing a noisy estimate of that representation, but then iteratively refines it over the next to layers. (c) A classic group of three layers can also distribute the computation, but they would produce a new representation at each layer. The iterative estimation stage in (b) can be seen as a middle ground between a single classic neural network layer, (a), and multiple classic layers, (c).

and changed between layers—an observation that contradicts the idea of building a new level of abstraction at each layer.

**Lesioning.**    If it were true that each layer computes a completely new set of features, then removing a layer from a trained network would completely change the input distribution for the next layer. We would then expect to see the overall performance drop to almost chance level. This is in fact what Veit et al. (2016) find for the 15-layer VGG network on CIFAR-10: removing any layer from the trained network sets the classification error to around 90%. But the lesioning studies conducted on Highway networks (Srivastava et al., 2015a) and ResNets (Veit et al., 2016) paint an entirely different picture: only a minor drop in performance is observed for any removed layer. This drop is more pronounced for the early layers and the layers that change dimensionality (i.e. number of filter maps and map sizes), but performance is always still far superior to random guessing.

Huang et al. (2016b) take lesioning one step further and drop out entire ResNet layers as a regularizer during training. They describe their method as "[...] a training procedure that enables the seemingly contradictory setup to *train short* networks and *use deep* networks at test time". The regularization effect of this procedure is explained as inducing an implicit ensemble of many shallow networks akin to normal dropout. Note that this explanation requires a departure from the representation view in that each layer has to cope with the possibility of having its entire input layer removed. Otherwise, most shallow networks in the ensemble would perform no better than chance level, just like the lesioned VGG net.

**Reshuffling.**    The link between layers and representation levels may be most clearly challenged by an experiment in Veit et al. (2016) where the layers of a trained 110-layer ResNet are reshuffled. Remarkably, error increases smoothly with the amount of reshuffling, and many re-orderings result only in a small increase in error. Note, however, that only layers within a stage are reshuffled, since the dimensionality of the swapped layers must match. Veit et al. (2016) take these results as evidence that ResNets behave as ensembles of exponentially many shallow networks.

## 3   UNROLLED ITERATIVE ESTIMATION VIEW

The representation view has guided neural networks research by providing intuitions about the "meaning" of their computations. In this section we will augment the representation view to deal with the incongruities and hopefully enable future research on these very deep architectures to reap the same benefits. The target of our modification is the mapping of layers/blocks of the network to levels of abstraction.

At this point it is interesting to note that the one-to-one mapping of neural network layers to levels of abstraction is an implicit assumption rather than a stated part of the representation view. A recent deep learning textbook (Goodfellow et al., 2016) explicitly states: "[. . . ] the depth flowchart of the computations needed to compute the representation of each concept may be much deeper than the graph of the concepts themselves." So in a strict sense the evidence from Section 2 does not in fact contradict a representation view of Residual and Highway networks. It only conflicts with the idea

that each layer forms a new level of representation. We can therefore reconcile very deep networks with the representation view by explicitly giving up this assumption.

**Unrolled Iterative Estimation.** We propose to think of blocks in Highway and Residual networks as performing *unrolled iterative estimation* of representations. By that we mean that the blocks in a stage work together to estimate and iteratively refine a single level of representation. The first layer in that stage already provides a (rough) estimate for the final representation. Subsequent layer in the stage then refine that estimate without changing the level of representation. So if the first layer in a stage detects simple shapes, then the rest of the layers in that stage will work at that level too.

A good initial estimate for a representation should on average be correct even though it might have high variance. We can thus formalize the notion of "preserving feature identity" as being an unbiased estimator for the target representation. This means the units $a_i^k$ in different layers $k \in \{1 \ldots L\}$ are all estimators for the same latent feature $A_i$, where $A_i$ refers to the (unknown) value towards which the $i$-th feature is converging. The unbiased estimator condition can then be written as the expected difference between the estimator and the final feature:

$$\mathbb{E}_{\mathbf{x} \in \mathbf{X}}[a_i^k - A_i] = 0. \tag{1}$$

Note that both the $a_i^k$s and $A_i$ depend on the samples $\mathbf{x}$ of the data-generating distribution $\mathbf{X}$ and are thus random variables. The fact that they both depend on the same $\mathbf{x}$ is also the reason we need to keep them within the same expectation and cannot just write $\mathbb{E}[a_i^k] = A_i$.

**Feature Identity.** A stage that performs iterative estimation is different from one that computes a new level of representation at each block because it *preserves the feature identity*. They operate differently even if their structure and their final representations are equivalent, because of the way they treat intermediate representations. This is illustrated in Figure 2, where the iterative estimation stage, (b), is contrasted with a single classic block (a), and multiple classic blocks, (c). In the iterative estimation case (middle), all the blocks within the stage produce estimates of the same representation (indicated by having different shades of blue). Whereas, in a classical stage, (c), the intermediate representations would all be different (represented by different colors).

## 3.1 HIGHWAY AND RESIDUAL NETWORKS

Both Highway and Residual networks address the problem of training very deep architectures by improving the error flow via identity skip connections that allow units to copy their inputs on to the next layer unchanged. This design principle was originally introduced in Long Short-Term Memory (LSTM) recurrent networks (Hochreiter & Schmidhuber, 1997) and mathematically these architectures correspond to a simplified LSTM network, "unrolled" over time.

In Highway Networks, for each unit there are two additional gating units, which control how much (typically non-linear) transformation is applied (transform gate $T$) and how much to just copy of the activation from the corresponding unit in the previous layer (carry gate $C$). Let $H(\mathbf{x})$ be a nonlinear parametric function of the inputs, $\mathbf{x}$, (typically an affine projection followed by pointwise non-linearity). Then a traditional feed-forward network layer can be written as:

$$y(\mathbf{x}) = H(\mathbf{x}). \tag{2}$$

By adding two additional units, $T(\mathbf{x})$ and $C(\mathbf{x})$ a Highway layer can be written as:

$$y(\mathbf{x}) = H(\mathbf{x}) \cdot T(\mathbf{x}) + \mathbf{x} \cdot C(\mathbf{x}). \tag{3}$$

Usually this is further simplified by coupling the gates, i.e. setting $C(\mathbf{x}) = 1 - T(\mathbf{x})$:

$$y(\mathbf{x}) = H(\mathbf{x}) \cdot T(\mathbf{x}) + \mathbf{x} \cdot (1 - T(\mathbf{x})). \tag{4}$$

ResNets simplify the Highway networks approach by reformulating the desired transformation as the input plus a residual $F(\mathbf{x})$. The rationale behind this is that it is easier to optimize the residual form than the original function. For the extreme case where the desired function is the identity, this amounts to the trivial task of pushing the residual to zero:

$$y(\mathbf{x}) = F(\mathbf{x}) + \mathbf{x}. \tag{5}$$

As with Highway networks, Residual networks can be viewed as unfolded recurrent neural networks of the particular mathematical form (one with an identity self-connection) of an LSTM cell. This has been explicitly pointed out by Liao & Poggio (2016), who also argue that this could allow Residual networks to emulate recurrent processing in the visual cortex and thus adds to their biological plausibility. Setting $F(\mathbf{x}) = T(\mathbf{x})[H(\mathbf{x}) - \mathbf{x}]$ converts Equation 5 to Equation 4 showing that both formulations differ only in the precise functional form for $F$. Alternatively, Residual networks can be seen as a particular case of Highway networks where $C(\mathbf{x}) = T(\mathbf{x}) = \mathbf{1}$ and are not learned.

## 3.2 DERIVING RESIDUAL NETWORKS

Equation 1 can be used to directly derive the ResNet equation (Equation 5). First, it follows that the expected difference between outputs of two consecutive blocks in a stage is zero:

$$\mathbb{E}[a_i^k - A_i] - \mathbb{E}[a_i^{k-1} - A_i] = 0 \tag{6}$$

$$\mathbb{E}[a_i^k - a_i^{k-1}] = 0. \tag{7}$$

If we write feature $a_i^k$ as a combination of $a_i^{k-1}$ and a residual $F_i$, it follows from Equation 7 that the residual has to be zero-mean:

$$a_i^k = a_i^{k-1} + F_i \tag{8}$$

$$\implies \mathbb{E}[F_i] = 0. \tag{9}$$

Therefore, if the residual block $F$ has a zero mean over the training set, then Equation 1 holds and it can be said to maintain feature identity. Note that this is a reasonable assumption, especially when using batch normalization.

## 3.3 DERIVING HIGHWAY NETWORKS

The coupled Highway formula (Equation 4) can be directly derived as an alternative way of ensuring Equation 1 if we assume a $H_i$ to be a new estimate of $A_i$. Highway layers then result from the optimal way to linearly combine the former estimate $a_i^{k-1}$ with $H_i$ such that the resulting $a_i^k$ is a minimum variance estimate of $A_i$, i.e. requiring $\mathbb{E}[a_i^k - A_i] = 0$ and that $\mathrm{Var}[a_i^k - A_i]$ is minimal.

Let $\alpha_1 = \mathrm{Var}[a_i^k - A_i] - \mathrm{Cov}[a_i^k - A_i, a_i^k - H_i]$ and $\alpha_2 = \mathrm{Var}[H_i - A_i] - \mathrm{Cov}[a_i^k - A_i, a_i^k - H_i]$, then the optimal linear way of combining them is then given by the following estimator (see Section A.1 for derivation):

$$a_i^{k+1} = \frac{\alpha_2}{\alpha_1 + \alpha_2} a_i^k + \frac{\alpha_1}{\alpha_1 + \alpha_2} H_i. \tag{10}$$

If we use a neural network to compute $H_i$ and another one to compute $T_i = \frac{\alpha_1}{\alpha_1 + \alpha_2}$, then we recover the Highway formula:

$$a_i^k = H_i \cdot T_i + a_i^{k-1} \cdot (1 - T_i), \tag{11}$$

where $H_i$ and $T_i$ are both functions of the previous layer activations $\boldsymbol{a}^{k-1}$.

# 4 DISCUSSION

## 4.1 IMPLICATIONS FOR HIGHWAY NETWORKS

In Highway networks with coupled gates the mixing coefficients always sum to one. This ensures that the expectation of the new estimate will always be correct (cf. Equation 14). The precise value of mixing will only determine the variance of the new estimate. We can bound this variance to be less or equal to the variance of the previous layer by restricting both mixing coefficients to be positive. In Highway networks this is done by using the logistic sigmoid activation function for the transform gate $T_i$. This restriction is equivalent to the assumption of $\alpha_1$ and $\alpha_2$ having the same sign. This assumption holds, for example, if the error of the new estimate $H_i - A_i$ is independent of the old $a_i^{k-1} - A_i$. Because in that case their covariance is zero and thus both alphas are positive.

Using the logistic sigmoid as activation function for the transform gate further means that the pre-activation of $T_i$ implicitly estimates $\log(\frac{\alpha_2}{\alpha_1})$. This is easy to see because the logistic sigmoid of that

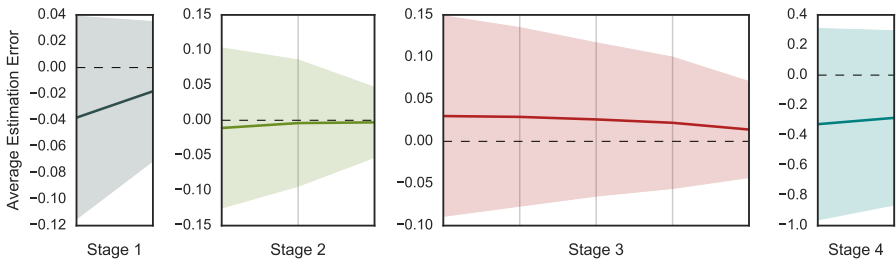

Figure 3: Experimental corroboration of Equation 1. The average estimation error – an empirical estimate of the LHS in Equation 1 – for each block of each stage (x-axis). It stays close to zero in all stages of a 50-layer ResNet trained on the ILSVRC-2015 dataset. The standard deviation of the estimation error decreases as depth increases in each stage (left to right), indicating iterative refinement of the representations.

term is

$$\frac{1}{1 + e^{\log(\frac{\alpha_2}{\alpha_1})}} = \frac{1}{1 + \frac{\alpha_2}{\alpha_1}} = \frac{\alpha_1}{\alpha_1 + \alpha_2}. \tag{12}$$

For the simple case of independent estimates ($\text{Cov}[a_i^k - A_i, a_i^k - H_i] = 0$), this gives us another way of understanding the transform gate bias: It controls our initial belief in the variance of the layers estimate as compared to the previous one. A low bias means that the layers on average produce a high variance estimate, and should thus only contribute little, which seems a reasonable assumption for initialization.

## 4.2 EXPERIMENTAL CORROBORATION OF ITERATIVE ESTIMATION VIEW

The primary prediction of the iterative estimation view is that the estimation error for Highway or Residual blocks within the same stage should be zero in expectation. To empirically test this claim, we extract the intermediate layer outputs for 5000 validation set images using the 50-layer ResNet trained on the ILSVRC-2015 dataset from He et al. (2015). These are then used to compute the empirical mean and standard deviation of the estimation error over the validation subset, for all blocks in the four Residual stages in the network. Finally the mean of the empirical mean and standard deviation is computed over the three spatial dimensions.

Figure 3 shows that for the first three stages, the mean estimation error is indeed close to zero. This indicates that it is valid to interpret the role of Residual blocks in this network as that of iteratively refining a representation. Moreover, in each stage the standard deviation of the estimation error decreases over successive blocks, indicating the convergence of the refinement procedure. We note that stage four (with three blocks) appears to be underestimating the representation values, indicating a probable weak link in the architecture.

## 4.3 VISUAL EVIDENCE & STAGE-WISE ESTIMATION OF FEATURES

ResNets (He et al., 2015) and many other derived architectures share some common characteristics: They are divided into stages of Residual blocks that share the same dimensionality. In between these stages the input dimensionality changes, typically by down-sampling and an increase in the number of channels. These stages typically also increase in length: the early stages consist of fewer layers compared to later ones.

We can now interpret these design choices from an iterative estimation point of view. From this perspective *the level of representation stays the same within each stage*, through the use of identity shortcut connections. Between stages, the level of representation is changed by the use of a projection to change dimensionality. This means that we expect the type of features that are detected to be very similar within a stage and jump in abstraction between stages. This view also suggests that the first few stages can be shorter, since low level representations tend to be relatively simple and need little

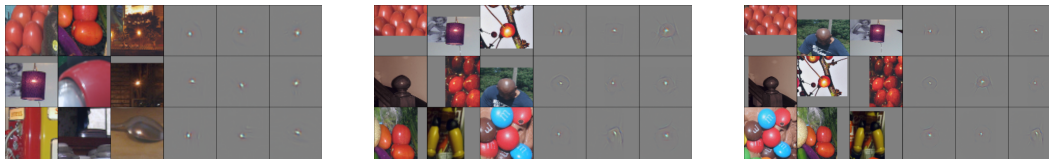

Figure 4: Feature visualization from Chu et al. (2017), reproduced with kind permission of the authors. It shows how the response of a single filter (unit) evolves over the three blocks (shown from left to right) of stage 1 in a 50-layer ResNet trained on ImageNet. On the left of each visualization are the top 9 patches from the ImageNet validation set that maximally activated that filter. To the right the corresponding guided backpropagation (Springenberg et al., 2014) visualizations are shown.

iterative refinement. The features of later stages on the other hand are likely complex with numerous inter-dependencies and therefore benefit more from iterative refinement.

Many visualization studies (such as those by Zeiler & Fergus (2014)) have examined the activities in trained convolutional networks and found evidence supporting the representation view. However, these studies were conducted on networks not designed for iterative estimation. The interpretation above paints a different picture for networks which learn unrolled iterative estimation. In these networks, we should observe stages and not layers corresponding to levels of representation.

Indeed, visualization of Residual network features supports the iterative estimation view. In Figure 4 we reproduce visualizations from a study by Chu et al. (2017) who observe: "[. . . ] residual layers of the same dimensionality learn features that get refined and sharpened". These visualizations show how the response of a single filter changes over three Residual blocks within the same stage of a 50-layer Residual network trained for image classification. Note that the filter appears to refine its response by including surrounding context, rather than changing it across blocks in the same stage. In the first block, the top nine activating patches for the filter include three light sources and six specular highlights. In later blocks, through the incorporation of spatial context, eight out of nine maximally activating patches are specular highlights. Similar refinement behavior is observed throughout the different stages of the network.

Another finding in line with this implication of the iterative estimation view is that in some cases sharing weights of the Residual blocks within a stage doesn't deteriorate performance much (Liao & Poggio, 2016). Similarly Lu & Renals (2015) shared the weights of the transform and carry gates of a thin and deep highway network, while still achieving better performance than both normal deep neural networks and Residual networks.

## 4.4 Revisiting Evidence against the Representation View

**Staying Close to the Inputs.** When iteratively re-estimating a variable, staying close to the old value should be a more common operation than changing it significantly. This is the reason why the ResNet formulation makes sense: learning the identity is hard *and* it is needed frequently. It also explains sparse transform gate activity in trained Highway networks: These networks learn to dynamically and selectively update individual features, while keeping most of the representation intact.

**Lesioning.** Another implication of the iteration view is that processing in layers is incremental and somewhat interchangeable. Each layer (apart from the first) refines an already reasonable estimate of the representation. It follows that removing layers, like in the lesioning experiments, should have only a mild effect on the final result because doing so does not change the overall representation the next layer receives, only its quality. The following layer can still perform mostly the same operation, even with a somewhat noisy input. Layer dropout (Huang et al., 2016b) amplifies this effect by explicitly training the network to work with a variable number of iterations. By dropping random layers it further penalizes iterations relying on each other, which could be another explanation for the regularization effect of the technique.

**Shuffling.** The layers within a stage should also be interchangeable to a certain degree, because they all work with the same input and output representations. Of course, this interchangeability is

| Variant | Functional Form | Perplexity |
|---------|-----------------|------------|
| Plain | $H(\mathbf{x})$ | 92.60 |
| Residual | $H(\mathbf{x}) + x$ | 91.32 |
| T-Only | $H(\mathbf{x}) \cdot T(\mathbf{x}) + \mathbf{x}$ | 82.94 |
| C-Only | $H(\mathbf{x}) + \mathbf{x} \cdot C(\mathbf{x})$ | 79.15 |
| Coupled | $H(\mathbf{x}) \cdot T(\mathbf{x}) + \mathbf{x} \cdot (1 - T(\mathbf{x}))$ | 79.13 |
| Full | $H(\mathbf{x}) \cdot T(\mathbf{x}) + \mathbf{x} \cdot C(\mathbf{x})$ | 79.09 |

(a) Comparing of various variants of the Highway formulation for character-aware neural language models (Kim et al., 2015).

| Variant | Top5 Error |
|---------|------------|
| Highway | $10.03 \pm 0.17$ |
| Highway-Full | $10.21 \pm 0.03$ |
| Resnet | $9.40 \pm 0.18$ |
| Highway + BN | $7.53 \pm 0.05$ |
| Highway-Full + BN | $7.29 \pm 0.11$ |
| Resnet + BN | $7.17 \pm 0.14$ |

(b) Comparing ILSVRC-2012 top5 classification error. Mean and std over 3 runs.

Table 1: Comparison of several Highway network and Residual network variants.

not without limitations. The network could learn to depend on a specific order of refinements, which would be disturbed by shuffling and lesioning. But we can expect these effects to be moderate in many cases, which is indeed what has been reported in the literature.

## 5 COMPARATIVE CASE STUDIES

The preceding sections show that we can construct both Highway and Residual architectures mathematically grounded in learning unrolled iterative estimation. The common feature between these architectures is that they preserve feature identities, and the primary difference is that they have different biases towards switching feature identities. Unfortunately, since our current understanding of the computations required to solve complex problems is limited, it is extremely hard to say *a priori* which architecture may be more suitable for which type of problems. Therefore, in this section we perform two case studies comparing and contrasting their behavior experimentally. The studies are each based on applications for which Residual and Highway layers respectively have been effective.

### 5.1 IMAGE CLASSIFICATION

Deep Residual networks outperformed all other entries at the 2016 ImageNet classification challenge. In this study we compare the performance of 50-layer convolutional Highway and Residual networks for ImageNet classification. Our aim is not to examine the importance of depth for this task— shallower networks have already outperformed deep Residual networks on all original Residual network benchmarks (Huang et al., 2016a; Szegedy et al., 2016). Instead, our goal is to fairly compare the two architectures, and test the following claims regarding deep convolutional Highway networks (He et al., 2015; 2016; Veit et al., 2016):

1. They are harder to train, leading to stalled training or poor results.
2. They require extensive tuning of the initial bias, and even then produce much worse results compared to Residual networks.
3. They are wasteful in terms of parameters since they utilize extra learned gates, doubling the total parameters *for the same number of units* compared to a Residual layer.

We train a 50-layer convolutional Highway network based on the 50-layer Residual network from He et al. (2015). The design of the two networks are identical (including use of batch normalization (BN) after every convolution operation), except that unlike Residual blocks, the Highway blocks use two sets of layers to learn $H$ and $T$ and then combine them using the coupled Highway formulation. We train two slight variations of the Highway network: *Highway*, in which $H$ has the same design as in a Residual block before addition i.e. `Conv-BN-ReLU-Conv-BN-ReLU-Conv-BN`, and *Highway-Full*, in which an additional third `ReLU` operation is added. The design of $T$ is `Conv-BN-ReLU-Conv-BN-ReLU-Conv-BN-Sigmoid`. As proposed initially for Highway layers, both $H$ and $T$ are learned using the same receptive fields and number of parameters. The transform gate biases are set to $-1$ at the start of training. For fair comparison, the number of feature maps throughout the Highway network is reduced such that the total number of parameters is close to the Residual network. The training algorithm and learning rate schedule are kept the same as those used for the Residual network.

The plots in Figure 5a show that the Residual network fits the data better—its final training loss is lower than the Highway network. The final performance of both networks on the validation set (see Table 1b) is very similar, with the Residual network producing a slightly better top-5 classification error of 7.17% vs. 7.53% for the *Highway* network. The *Highway-Full* network produces even closer results with a mean error of 7.29%. *These results contradict claims 1 and 2 above*, since the Highway networks are easy to train without requiring any bias tuning. However, there is some support for claim 3 since the Highway network appears to slightly underfit compared to the Residual network, suggesting lower capacity for the same number of parameters.

**Importance of Expressive Gating.**    The mismatch between the results above and claims 1 and 2 made by He et al. (2016) can be explained based on the importance of having sufficiently expressive transform gates. For experiments with Highway networks (which they refer to as *Residual networks with exclusive gating*), He et al. (2016) used $1 \times 1$ convolutions for the transform gate, instead of having the same receptive fields for the gates as the primary transformation ($H$), as done by Srivastava et al. (2015a). This change in design appears to be the primary cause of instabilities in learning since the gates can no longer function effectively. Therefore, it is important to use equally expressive transformations for $H$ and $T$ in Highway networks.

**Role of Batch Normalization.**    Since both architectures have built-in ease of optimization compared to plain networks, it is interesting to investigate the necessity of batch normalization for training these networks. Our derivation in Section 3.2 suggest that BN in Residual networks could take the role of an inductive bias towards iterative estimation by keeping the expected mean of the residual zero (cf. Equation 9). To investigate its role we train the networks above without any batch normalization. The resulting training curves are shown in Figure 5b of the supplementary.

We find that without BN both networks reach an even lower training error than before while performing worse on the validation set indicating increased overfitting for both. This shows that BN is not necessary for training these networks and does not speed up learning. Interestingly, the effect is more pronounced for the Highway network, which now fits the data better than the ResNet. *This contradicts claim 3*, since a Highway network with the same number of parameters as a Residual network demonstrates slightly higher capacity. On the other hand both networks produce a higher validation error—10.03% and 9.40% for the Highway and Residual network respectively—indicating a clear case of overfitting. This means that batch normalization provides regularization benefits that can't easily be explained by either improved optimization nor by the inductive bias for Residual networks.

## 5.2    LANGUAGE MODELING

Next we compare different functional forms (or *variants*) of the Highway network formulation for the case of character-aware language modeling. Kim et al. (2015) have shown that utilizing a few Highway fully connected layers instead of conventional *plain* layers improves model performance for a variety of languages. The architecture consists of a stack of convolutional layers followed by Highway layers and then an LSTM layer which predicts the next word based on the history. Similar architectures have since been utilized for obtaining substantial improvements for large-scale language modeling (Jozefowicz et al., 2016) and character level machine translation (Lee et al., 2016). Highway layers with coupled gates have been used in all these studies.

Only two to four Highway layers were necessary to obtain significant modeling improvements in the studies above. Thus, it is reasonable to assume that the central advantage of using Highway layers for this task is not easing of credit assignment over depth, but an improved modeling bias. To test how well Residual and other variants of Highway networks perform, we compare several language models trained on the Penn Treebank dataset using the same setup and code provided by Kim et al. (2015). We use the LSTM-Char-Large model, only changing the two Highway layers to different variants. The following variants are tested:

**Full**  The original Highway formulation based on the LSTM cell. We note that this variant uses more parameters than the others, since changing the layer size to reduce parameters would affect the rest of the network architecture as well.

**Coupled**  The most commonly used Highway variant, derived in Section 3.3.

**C-Only**  A Highway variant with a carry gate but no transform gate (always set to one).

**T-Only**  A Highway variant with a transform gate but no carry gate (always set to one).

**Residual**  The Residual form from He et al. (2015), in which both transform and carry gate are always one. For this variant we use four layers instead of two, to match the amount of computation/parameters of the other variants.

The test set perplexity of each model is shown in Table 1a. We find that the the Full, Coupled and C-Only variants have similar performance, better than the T-Only variant and substantially better than the Residual variant. The Residual variant results in performance close to that obtained by using a single plain layer, even though four Residual layers are used. Learned gating of the identity connection is crucial for improving performance for this task.

Recall that the Highway layers transform character-aware representations before feeding them into an LSTM layer. Thus the non-contextual word-level representations resulting from the convolutional layers are transformed into representations better suited for contextual language modeling. Since it is unlikely that the entire representation needs to change completely, this setting fits well with the iterative estimation perspective.

Interestingly, Table 1a shows a significant advantage for all variants with a multiplicative gate on the inputs. These results suggest that in this setting it is crucial to dynamically replace parts of the input representation. Some features need to be changed drastically *conditioned on other detected features* such as word type while other features need to be retained. As a result, even though Residual networks are compatible with iterative estimation, they may not be the best choice for tasks where mixing adaptive feature transform/replacement and reuse is required.

## 6  CONCLUSION

This paper offers a new perspective on Highway and Residual networks as performing unrolled iterative estimation. As an extension of the popular representation view, it stands in contrast to the optimization perspective from which these architectures have originally been introduced. According to the new view, successive layers (within a stage) cooperate to compute a single level of representation. Therefore, the first layer already computes a rough estimate of that representation, which is then iteratively refined by the successive layers. Unlike layers in a conventional neural network, which each compute a new representation, these layers therefore *preserve feature identity*.

We have further shown that both Residual and Highway networks can be directly derived from this new perspective. This offers a unified theory from which these architectures can be understood as two approaches to the same problem. This view further provides a framework from which to understand several surprising recent findings like resilience to lesioning, benefits of layer dropout, and the mild negative effects of layer reshuffling. Together with the derivations these results serve as compelling evidence for the validity of our new perspective.

Motivated by their conceptual similarities we set out to compare Highway and Residual networks. In preliminary experiments we found that they give very similar results for networks of equal size, thus refuting some claims that Highway networks would need more parameters, or that any form of gating impairs the performance of Residual networks. In another example, we found non-gated identity skip-connections to perform significantly worse, and offered a possible explanation: If the task requires dynamically replacing individual features, then the use of gating is beneficial.

The preliminary evidence presented in this report is meant as a starting point for further investigation. We hope that the unrolled iterative estimation perspective will provide valuable intuitions to help guide research into understanding, improving and possibly combining these exciting techniques.

ACKNOWLEDGEMENTS

The authors wish to thank Faustino Gomez, Bas Steunebrink, Jonathan Masci, Sjoerd van Steenkiste and Christian Osendorfer for their feedback and support. We are grateful to NVIDIA Corporation for providing us a DGX-1 as part of the Pioneers of AI Research award. This research was supported by the EU project "INPUT" (H2020-ICT-2015 grant no. 687795).

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

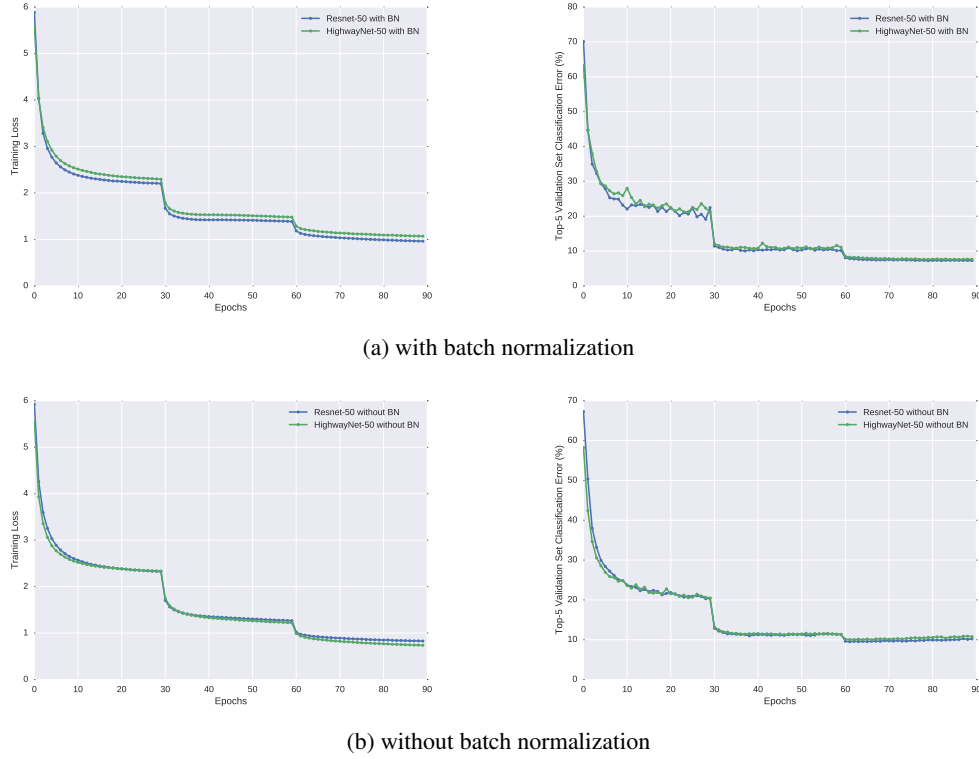

(a) with batch normalization

(b) without batch normalization

Figure 5: Comparing 50-layer Highway vs. Residual networks on ILSVRC-2012 classification.

## A    DERIVATION

### A.1    OPTIMAL LINEAR ESTIMATOR

Assume two random variables $A$ and $B$ that are both noisy measurements of a third (latent) random variable $C$:

$$\mathbb{E}[A - C] = \mathbb{E}[B - C] = 0 \tag{13}$$

We call the corresponding variances $\text{Var}[A - C] = \sigma_A^2$ and $\text{Var}[B - C] = \sigma_B^2$ and covariance $\text{Cov}[A, B] = \sigma_{AB}^2$.

We are looking for the linear estimator $q(A, B) = q_0 + q_1 A + q_2 B$ of $C$ with $\mathbb{E}[q - C] = 0$ (unbiased) that has minimum variance.

$$\mathbb{E}[q(A, B) - C] = 0$$
$$\mathbb{E}[q_0 + q_1 A + q_2 B - C] = 0$$
$$\mathbb{E}[q_0 + q_1 A - q_1 C + q_2 B - q_2 C + (q_1 + q_2 - 1)C] = 0$$
$$\mathbb{E}[q_0 + q_1(A - C) + q_2(B - C) + (q_1 + q_2 - 1)C] = 0$$
$$q_0 + (q_1 + q_2 - 1)\mathbb{E}[C] = 0$$
$$\mathbb{E}[C](1 - q_1 - q_2) = q_0$$

for all $\mathbb{E}[C]$ which is possible iff:

$$q_0 = 0 \text{ and } q_1 + q_2 = 1. \tag{14}$$

The second condition about minimal variance thus reduces to:

$$\underset{q_1, q_2}{\text{minimize}} \quad \text{Var}[q_1 A + q_2 B - C]$$
$$\text{subject to} \quad q_1 + q_2 = 1$$

We can solve this using Lagrangian multipliers. For that we need to take the derivative of the following term w.r.t. $q_1$, $q_2$ and $\lambda$ and set them to zero:

$$\text{Var}[q_1 A + q_2 B - C] - \lambda(q_1 + q_2 - 1))$$

The first equation is therefore:

$$\frac{d}{dq_1}\left(\text{Var}[q_1 A + q_2 B - C] - \lambda(q_1 + q_2 - 1)\right) = 0$$

$$\frac{d}{dq_1}\text{Var}[q_1 A + q_2 B - C] - \lambda = 0$$

$$\frac{d}{dq_1}\text{Var}[q_1(A - C) + q_2(B - C)] - \lambda = 0$$

$$\frac{d}{dq_1}\left(q_1^2\,\text{Var}[A - C] + 2q_1 q_2\,\text{Cov}[A - C, B - C]\right) - \lambda = 0$$

$$2q_1\sigma_A^2 + 2q_2\sigma_{AB}^2 - \lambda = 0$$

Analogously we get:

$$2q_2\sigma_B^2 + 2q_1\sigma_{AB}^2 - \lambda = 0$$

and:

$$q_1 + q_2 = 1$$

Solving these equations gives us:

$$q_1 = \frac{\sigma_B^2 - \sigma_{AB}^2}{\sigma_A^2 - 2\sigma_{AB}^2 + \sigma_B^2} \tag{15}$$

$$q_2 = \frac{\sigma_A^2 - \sigma_{AB}^2}{\sigma_A^2 - 2\sigma_{AB}^2 + \sigma_B^2} \tag{16}$$

$$\tag{17}$$

We can write our estimator in terms of $\alpha_1 = \sigma_B^2 - \sigma_{AB}^2$ and $\alpha_2 = \sigma_A^2 - \sigma_{AB}^2$:

$$q = \frac{\alpha_1}{\alpha_1 + \alpha_2}A + \frac{\alpha_2}{\alpha_1 + \alpha_2}B$$

