# Peer review of "Highway and Residual Networks learn Unrolled Iterative Estimation"

_ICLR 2017 — accepted_

[Official Review · AnonReviewer2 · rating 8 · confidence 4 · 15 Dec 2016]
**An interesting perspective on representations in DNNs**

The paper describes an alternative view on hierarchical feature representations in deep neural networks. The viewpoint of refining representations is well motivated and is in agreement with the success of recent model structures like ResNets.

Pros:

- Good motivation for the effectiveness of ResNets and Highway networks
- Convincing analysis and evaluation

Cons:

- The effect of this finding of the interpretation of batch-normalization is only captured briefly but seems to be significant
- Explanation of findings in (Zeiler & Fergus (2014)) using UIE viewpoint missing

Remarks:

- Missing word in line 223: "that it *is* valid"

[Official Review · AnonReviewer3 · rating 6 · confidence 5 · 15 Dec 2016]
**A New Perspective of ResNet and Highway Network**

This paper provides a new perspective to understanding the ResNet and Highway net. The new perspective assumes that the blocks inside the networks with residual or skip-connection are groups of successive layers with the same hidden size, which performs to iteratively refine their estimates of the same feature instead of generate new representations. Under this perspective, some contradictories with the traditional representation view induced by ResNet and Highway network and other paper can be well explained.

The pros of the paper are:
1. A novel perspective to understand the recent progress of neural network is proposed.
2. The paper provides a quantitatively experimentals to compare ResNet and Highway net, and shows contradict results with several claims from previous work. The authors also give discussions and explanations about the contradictories, which provides a good insight of the disadvantages and advantages between these two kind of networks.

The main cons of the paper is that the experiments are not sufficient. For example, since the main contribution of the paper is to propose the “unrolled iterative estimation" and the stage 4 of Figure 3 seems not follow the assumption of "unrolled iterative estimation" and the authors says: "We note that stage four (with three blocks) appears to be underestimating the representation values, indicating a probable weak link in the architecture.". Thus, it would be much better to do experiments to show that under some condition, the performance of stage 4 can follow the assumption. 

Moreover, the paper should provide more experiments to show the evidence of "unrolled iterative estimation", not comparing ResNet with Highway Net. The lack of experiments on this point is the main concern from myself.

[Official Review · AnonReviewer1 · rating 7 · confidence 4 · 22 Dec 2016]
**An interesting angle on resNets and Highway nets**

Thank you for an interesting angle on highway and residual networks. This paper shows a new angle to how and what kind of representations are learnt at each layer in the aforementioned models. Due to residual information being provided at a periodic number of steps, each of the layers preserve feature identity which prevents lesioning unlike convolutional neural nets.                                 
                                                                                                                                                                                                          
Pros                                                                                                                                                                                                      
- the iterative unrolling view was extremely simple and intuitive, which was supported by theoretical results and reasonable assumptions.                                                                 
- Figure 3 gave a clear visualization for the iterative unrolling view                                                                                                                                    
                                                                                                                                                                                                          
Cons                                                                                                                                                                                                      
- Even though, the perspective is interesting few empirical results were shown to support the argument. The major experiments are image classification and language models trained on mutations of character-aware neural language models.                                                                                                                                                                         
- Figure 4 and 5 could be combined and enlarged to show the effects of batch normalization.

[Author Response · Klaus Greff · 18 Jan 2017]
**Updated Version**

We've uploaded an updated version of the paper which addresses reviewers' concerns and makes several improvements. Apologies for the late revision, mainly due to the time-consuming ImageNet experiments. 

- To further confirm the iterative estimation, and contrast our view with the findings of Zeiler and Fergus (2014), we did a literature survey and planned on adding some visualizations of ResNet features.  We found a technical report from Brian Chu at Berkeley who applied known visualization techniques to Resnets independently and report exactly the behaviour we were expecting: The features within a stage get refined and sharpened. So now we refer to their paper and (with their kind permission) reproduce one of their visualizations. We have added a discussion of these findings which support our proposed view. 

- We've added mean+-std results for the experiments in Section 5.1.
These experiments take a long time, so we performed only three runs each, but the results are stable enough for comparison.

- We've elaborated a bit further on the role of batch normalization in Section 5.1. 

- improved language and organization

Apart from these changes we've also started investigating the unexpected behaviour of stage 4, by creating modified architectures. We've added three blocks to that stage and we've tried adding a fifth stage. We found that both variants improve performance to ca. 6.8% top5 error.
But the average estimation error stays high for the first few blocks and only later starts to decrease: [-0.32, -0.31, -0.30, -0.27, -0.17] (compare Figure 3)
We are further investigating this and we'll add our findings as soon as they paint a coherent picture.  But as of now, we didn't consider them to be interesting enough to be included in the paper.

[Final Decision · Program Chairs · 06 Feb 2017]
**ICLR committee final decision**

The paper provides interesting new interpretations of highway and residual networks, which should be of great interest to the community.